# Individual differences in self-reported lie detection abilities

**Mélanie Fernandes**[1]*, **Domicele Jonauskaite**[1,2], **Frédéric Tomas**[3], **Eric Laurent**[4], **Christine Mohr**[1]

**1** Institute of Psychology, University of Lausanne, Lausanne, Switzerland, **2** Faculty of Psychology, University of Vienna, Vienna, Austria, **3** Center for Cognition and Communication, University of Tilburg, Tilburg, Netherlands, **4** Department of Psychology, University of Franche-Comté, Besançon, France

\* melanie.fernandes@unil.ch

## Abstract

Previous literature on lie detection abilities bears an interesting paradox. On the group level, people detect others' lies at guessing level. However, when asked to evaluate their own abilities, people report being able to detect lies (i.e., self-reported lie detection). Understanding this paradox is important because decisions which rely on credibility assessment and deception detection can have serious implications (e.g., trust in others, legal issues). In two online studies, we tested whether individual differences account for variance in self-reported lie detection abilities. We assessed personality traits (Big-Six personality traits, Dark Triad), empathy, emotional intelligence, cultural values, trust level, social desirability, and belief in one's own lie detection abilities. In both studies, mean self-reported lie detection abilities were above chance level. Then, lower out-group trust and higher social desirability levels predicted higher self-reported lie detection abilities. These results suggest that social trust and norms shape our beliefs about our own lie detection abilities.

## Introduction

Evolutionary approaches highlight that living in social groups is key to being successful in our environments [1]. Living in groups comes not only with advantages though. Firstly, we must assume that other group members are trustworthy and bare no threat, so that we can engage in effective communication. Without trust, we would constantly experience problematic relationships within and between social groups. Interpersonal trust ensures individuals' well-being, the achievement of their own aims, and the longevity of social units such as families, communities, and institutions [2–4]. Yet, being excessively trustworthy also bears risks, because others might take advantage for their own benefits [4].

In an ideal world, one would always know who to trust and who not to trust, who and what to believe and not to believe. This ideal world, however, does not exist, because deception occurs daily [5, 6], on average, at least once or twice a day [7–9]. When people lie, they make completely or partially untrue statements with the intent to deceive, and about 80% of the times, people succeed with lies remaining undetected [7]. Yet, not everybody lies to the same extent with more recent studies showing that a few prolific liars are telling the majority of lies

**Data Availability Statement:** All the data files of the individual differences in self-reported lie detection abilities' study are available from the Open Science Framework database (DOI: 10. 17605/OSF.IO/6NRA5).

**Funding:** We thank the Swiss National Science Foundation for making this work possible (P500PS_202956, 100014_182138).

**Competing interests:** The authors have declared that no competing interests exist.

[6, 10, 11]. Then, there is not one type of lies. There are many: lies can be about feelings, opinions, achievements, failures, to name a few examples. We may lie because honesty would hamper social or material goals [7], for financial or emotional gains, to avoid embarrassment, or to protect the feelings of others [7, 12–14].

At this point, we wish to highlight a striking paradox between actual (objective) and self-reported (subjective) lie detection abilities. Experimental studies showed that objectively, people perform around chance level when trying to spot somebody else lying [15, 16]. Subjectively, however, people report being able to detect lies successfully [13, 17, 18]. This paradox might originate from the so-called *truth-default* state, meaning, one assumes that others are honest. Indeed, most everyday situations inspire trust, or at least, do not merit questioning. Such a *truth-default* state is desirable because trust facilitates cooperation and effective communication [19, 20]. Clare and Levine [19] argued that individuals would only deviate from this *truth-default* state subsequent to a trigger events, such as suspicious indicators or motives, dishonest behavior, information from third parties, or inconsistencies in statements or known facts.

Overall, we can assume that lying is omnipresent, but that most lies remain undetected, because people see few reasons to distrust each other. Consequently, individuals encounter few opportunities to study and learn about others' successful lies. Thus, we face an imbalance between i) the number of undetected lies by oneself, ii) the number of undetected lies by others, and iii) the number of a few occasions when lies were detected. This imbalance may leave individuals with an inflated estimation of their own lie detection abilities, once a lie had been detected [17]. Moreover, this inflated estimation might be further supported by people's tendency to think positively about themselves: being easily deceived would not match this positive self-view [21].

When trying to understand this inflated estimation, we noted a surprising void of published studies reporting on reasons or correlates of self-reported lie detection abilities [22–25]. The few available studies focused on the Dark Triad personality traits–Machiavellianism, narcissism, and psychopathy,–because these traits share malevolent behavioral tendencies such as self-promotion, manipulation (including lying), and lack of empathy [21–25]. Wissing and Reinhard [22] showed that higher psychopathy and higher overall scores on Dark Triad personality were associated with higher self-reported confidence in lie detection abilities. Then, Zvi and Elaad [23] showed that higher levels of narcissism were associated with higher production of lies, which in turn was associated with higher self-reported lie detection abilities in comparison to others [23]. The recent study by Elaad [25] supported the association between higher narcissism and higher self-reported lie detection abilities in comparison to others. Interestingly, self-reported confidence in one's own lie detection abilities was unrelated to individuals' actual lie detection abilities [26]. Also, variance in Dark Triad traits did not explain variance in actual lie detection abilities either [26].

Results on these few studies were not conclusive. Thus, we designed a new study on individual differences on self-reported lie detection abilities. In addition to the Dark Triad personality traits, we included further measures that had been studied in the context of lie detection abilities and lie production before, though again showing inconclusive results. To this end, we selected a series of different measures, namely gender, social cognition measures, cultural differences, general personality traits, and social desirability.

For gender, studies showed no differences between men and women in both lie detection and lie production [27–29]. For social cognition, we considered empathy and emotional intelligence. For instance, Duran et al. [30] found that women with lower as compared to higher levels of empathy performed better at lie detection. Former studies would have predicted the opposite, though [31, 32]. Others argued that higher levels of empathy should link to impaired lie detection performances [33, 34]. In respect to emotional intelligence, higher trait levels

might facilitate lie detection [35, 36]. Actual studies, however, showed that higher emotional intelligence, particularly emotionality, yielded impaired lie detection abilities [37].

For cultural differences, the need for lie detection might differ as a function of where you live on the globe, because levels of trust, deception, and corruption vary cross-culturally [4, 9, 38]. For instance, in case of lower national trust levels, people might be more suspicious towards others, having a lower *truth-default* state [20]. Maybe, they perform above chance in lie detection, or they just think that they do, once the odd lie had been detected. Moreover, individuals from collectivistic societies (prioritizing harmony among group members) seem more inclined to lie for other people's benefits than those from individualistic societies [39].

To account for individual differences, we performed two consecutive online studies to investigate self-reported lie detection abilities. In Study 1, participants completed self-report measures on the Dark Triad [40], empathy [41], cultural values [42], and national trust levels [43]. Participants also indicated whether they were able to detect lies (yes/no), and at which level of accuracy (0–25%, 25–50%, 50–75%, 75–100%). Around 80% of the sample answered *yes*, with little variance in the accuracy level ratings. Thus, we made several changes to Study 2. We again asked participants to complete some of the same questionnaires (empathy, cultural values, national trust levels), but exchanged the 12-item Dirty Dozen [40] with the psychometrically superior Dark Triad personality traits questionnaire [44]. Additionally, we assessed participants' general personality traits such as Agreeableness, Openness to Experience, Extraversion, Conscientiousness, Honesty and Resiliency [45], due to formerly described relationships with Dark Triad traits [46, 47]. We also assessed emotional intelligence [48] and socially desirable responding [49]. Crucially, participants again completed the yes/no question regarding their self-reported lie detection abilities. This time they rated their estimated accuracy on a scale from 0 (*unable to detect lies*) to 100 (*perfectly able to detect lies*). Moreover, to account for previous studies on the subject (see also [18, 23–25, 50, 51], we also asked participants to judge their lie detection abilities in comparison to others on a scale ranging from 0 (*much worse than others*) to 100 (*much better than others*).

Across these two studies, we investigated whether we might observe enhanced self-reported lie detection abilities as a function of i) Dark Triad personality scores (i.e., narcissism, psychopathy, and Machiavellianism), ii) lower empathy scores, iii) lower emotional intelligence scores, and iv) cultural values. Regarding cultural values, we expected to find differences between Hofstede's cultural values (i.e., collectivism) and self-reported lie detection abilities. Finally, we expected self-reported lie detection abilities to be positively associated with social desirability, in line with people's tendency to think and show themselves in a positive way [21].

## Study 1

### Method

**Participants.**   We recruited 764 participants (105 males). After excluding incomplete data and selecting participants between 18 and 31 years old (i.e., the majority), our final sample consisted in 487 participants (99 males) with a mean age of 21.50 years ($SD_{age}$ = 6.57 years; range = 18–31 years). Of these, 418 (95 males) were undergraduate students at the University of Lausanne, Switzerland, who received course credit for their participation. The remaining 69 participants (4 males) were recruited at the University of Franche-Comté, France. All participants were native French speakers. These studies were conducted in accordance with the principles expressed in the Declaration of Helsinki and received approval from the Research Ethics Commission of the Institute of Psychology, University of Lausanne (C_SSP_052021_00001).

We had used the statistical power analysis tool G*Power [52] to estimate the minimum sample size of 473 for a two-tailed binary logistic regression with a small effect size (odds ratio of 1.68) (see[53]), the $H0$ probability for $Y$ = 1 of .45, α of 0.05, power (1−β) of 0.80.

## Material

**Self-report questionnaires.** All our questionnaires demonstrated good or acceptable internal consistency (Cronbach's alpha values between .510 and .817) (see Table 1).

**Dark Triad Dirty Dozen inventory [54] (French-Canadian version from Savard et al. [40]).** This short 12-item scale assesses psychopathy, narcissism and Machiavellianism. Participants indicated their agreement with each statement on a 5-point Likert scale (see Table 1). Narcissism items assess the search for admiration and attention, the importance of status and expectations of special favours from others (e.g., "*I tend to want others to admire me*"). Machiavellianism items assess the use of deception, manipulation, flattery, and exploitation of others to achieve own goals (e.g., "*I tend to manipulate others to get my way*"). Psychopathy items assess the lack of remorse, insensitivity, being cynical, and being unconcerned by the morality of own actions (e.g., "*I tend to lack remorse*"). We averaged the scores of the four items of each subscale to obtain the total score on each subscale. Higher scores indicate higher expressions of narcissistic, Machiavellian, or psychopathic traits.

**Individual Cultural Values Scale [55] (French version from Zheng [42]).** This 26-item scale is based on Hofstede's cultural dimensions model [56, 57] assessing the five cultural dimensions of i) power distance, ii) uncertainty avoidance, iii) collectivism vs individualism iv) masculinity, and v) long-term vs. short-term orientation (see Table 1 for details on the scale and sub-scales). **Power distance** refers to the degree to which less powerful members of institutions and organizations expect and accept that power is unequally distributed. Higher scores indicate a stronger acceptance of this unequal power distribution. **Uncertainty avoidance** refers to the degree to which societies avoid uncertainty, are anxious, feel threatened by unknown situations, prefer clear procedures, and respect rules. Higher scores indicate a stronger uncertainty avoidance. **Collectivism vs. individualism** describes the degree to which collectivism (group cohesion and interests) dominates over individualism (one's own and close kins' interests). Lower scores indicate a bias towards collectivism, and higher scores–towards individualism. **Masculinity** describes the degree to which male vs. female role patterns dominate in a society, with higher scores favouring the male pattern. **Long-term vs. short term**

**Table 1. Description of the self-report questionnaires.**

| Construct and Scale | Subscales (items) | Response mode | Mean (SD) | Cronbach's α |
|---|---|---|---|---|
| **Dark personality traits (Dark Triad)** *The French-Canadian Dirty Dozen* | psychopathy (4) narcissism (4) Machiavellianism (4) | 1 (*strongly disagree*) to 5 (*strongly agree*) | 2.02 (0.33) 2.78 (0.46) 2.41 (0.49) | .596 .780 .747 |
| **Cultural dimensions** *Individual Cultural Values Scale* | power distance (5) uncertainty avoidance (5) collectivism (6) masculinity (4) long-term orientation (6) | 1 (*strongly disagree*) to 7 (*strongly agree*) 1 (*extremely unimportant to me*) to 7 (*extremely important to me*) | 1.84 (0.34) 5.12 (0.32) 3.27 (0.47) 2.14 (0.37) 5.35 (0.68) | .723 .817 .779 .660 .678 |
| **Trust level** *The World Value Survey 5* | In group trust level (3) Out group trust level (3) | 1 (*not at all*) to 4 (*completely*) | 3.11 (0.85) 2.64 (0.66) | .510 .710 |
| **Empathy** *Interpersonal Reactivity Index (IRI)* | Empathy score (28) | 0 (*does not describe me well*) to 4 (*describes me very well*) | 3.77 (0.79) | .814 |

Description of the self-report questionnaires, their subscales, and response mode. In addition, this table shows means, standard deviations (*SD*) and Cronbach's alpha values.

**orientation** refers to cultures that make long-term or short-term plans, respectively. The scale considers prudence, thrift, persistence, perseverance, and willingness to sacrifice the present to favour future successes. Higher scores indicate a preference for long-term planning.

Participants expressed their agreement with each statement on a 7-point Likert scale, and the scores were averaged for each subscale (see Table 1).

**The World Value Survey 5 [43].** This 6-item questionnaire determines **in-group** and **out-group trust** levels on a 4-point Likert scale (see Table 1). For the in-group trust level, participants rate their trust of family members, neighbours, and people they know personally. For out-group trust level, participants rate their trust of people they meet for the first time, from another religion, and another nationality. For the two subscales, the Likert scale scores were averaged (Table 1).

**Interpersonal Reactivity Index (IRI [58]; French version from Gilet et al. [41]).** The 28-item IRI questionnaire evaluates the following aspects of empathy: i) feelings of compassion and concern for unfortunate others, ii) the ability to consider the perspective of others, iii) participants' tendencies to transpose themselves imaginatively or to identify with fictional characters, and iv) self-oriented feelings of personal anxiety and distress in difficult interpersonal situations. Participants rated their agreement with items on a 5-point Likert scale. Of the 28 items, eight were reversely coded. We calculated an average empathy score so that higher scores indicated a natural capacity to share and understand the affective state of others [59] (see Table 1).

**Other questionnaires and measures.** In Study 1, we also assessed participants' self-reported trait autism scores [60] and paranormal belief scores [61]. In both studies, we also asked participants to report on cues they rely on to decide that another person is lying (open question added after the questions on self-reported lying detection abilities, Fig 1D). These data are reported elsewhere.

**Self-reported lie detection abilities.** We asked participants to indicate their self-reported lie detection abilities. In other words, we asked whether they were able to recognize somebody lying. Participants answered on a binary yes/no scale. Afterwards, participants indicated their self-reported lie detection abilities on a 4-point Likert-type scale. They had to rate how accurate they think they were at lie detection, choosing from four options: accurate 0–25% of the time (1), 25–50% (2), 50–75% (3), or 75–100% of the time (4).

## Procedure

We used the LimeSurvey platform to prepare and run our online survey. We distributed the online link to potential volunteers. In case of interest, they could complete the survey at their own convenience. On the first two pages, we provided, respectively, written study information

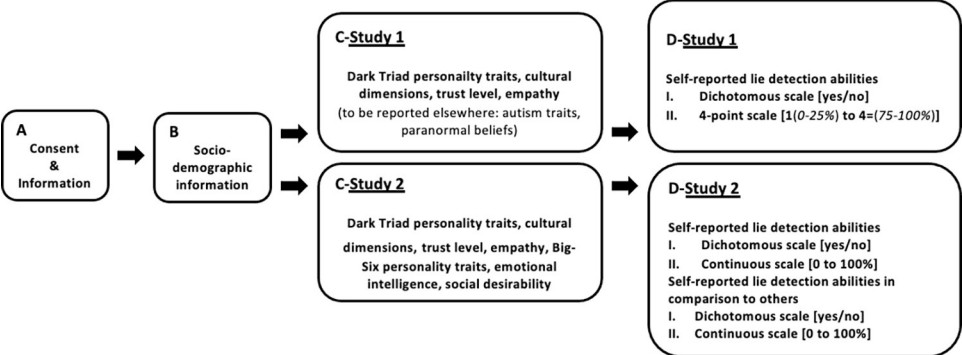

**Fig 1. Flow diagram of events.** The flow diagram depicts parts of the survey that i) were comparable to Study 1 and Study 2 (A, B), ii) were complemented by other questionnaires in Study 2 as compared to Study 1 (C), and iii) used different rating scales for self-reported lie detection abilities (D).

and ethical information, such as the right to withdraw from the study at any time, and data confidentiality (Fig 1A). We also stated that we treat participants' continuation as informed consent. Next, participants provided socio-demographic information regarding their age, gender, nationality, and field of studies (Fig 1B). Then, they completed the self-reported questionnaires in the following order: The FR-C Dark Triad Dirty Dozen, the Cultural Values Scale, the Interpersonal Reactivity Index, and their level of trust based on the World Value Survey 5 (Fig 1C-Study 1 and Table 1). Afterwards, they indicated their self-reported lie detection abilities (Fig 1D-Study 1). Finally, they were fully debriefed and thanked for their participation. The survey took about 20 minutes to complete.

**Design and statistical analysis.** To test for the role of individual differences in self-reported lie detection abilities, we considered the self-reported lie detection abilities score as a dependent measure, which was measured on a four-point scale (0–25%; 25–50%; 50–75%; and 75–100%). The histogram showed an uneven distribution. Of the 487 participants, 198 rated their abilities between 25–50% of accuracy and 237 between 50–75%, while only 19 participants self-reported 0–25% of accuracy, and only 33 indicate 75–100% of accuracy. Consequently, we created a dichotomous variable consisting in self-reported lie detection abilities that are below chance level (i.e. scores between 0–25% and 25–50% of accuracy) and above chance level (i.e. scores between 50–75% and 75–100% of accuracy).

To test our study question, we conducted a binary logistic regression analysis on this dependent dichotomous variable (below vs. above chance level) using the following 12 continuous predictor variables: psychopathy, narcissism, Machiavellianism, power distance, uncertainty avoidance, collectivism, masculinity, long-term orientation, empathy, ingroup trust level, outgroup trust level, and gender (see also S1 Table). We presented the correlations between all the predictor variables as supporting information (see S2 Table).

## Results

Overall, most participants (81.52%) reported being able to detect lies (yes/no answer) and many participants (55.44%) self-reported that their lie detection abilities ranged between 50% and 100% (i.e. above chance level self-reported lie detection abilities). The likelihood ratio test showed that the overall binary logistic regression model on self-reported lie detection abilities was not significant; $LR(12) = 656.23$, $p = .360$, $AIC = 682.23$, $_{pseudo}R^2 = .025$ (Cox & Snell), .034 (Nagelkerke) (see S1 Table).

## Study 2

### Method

**Participants.** We used the statistical power analysis tool G*Power [52] to estimate our sample size. We determined a minimum sample size of 231 for a linear multiple regression test with a small effect size of 0.1, 21 predictors, α of 0.05, and power (1−β) of 0.80.

We recruited a new sample of French speaking undergraduate psychology students ($N = 700$, 90 males) at the University of Lausanne. After excluding incomplete data and matching participants' age to those of Study 1, we were left with 386 participants (72 males; $M_{age} = 20.21$, $SD_{age} = 2.22$, range = 18 to 31 years). One academic year separated the data collection for Study 1 and Study 2. All participants received course credit for their participation.

### Material

**Self-report questionnaires. Short Dark Triad (SD3) [62] (French version from Gamache et al. [44]).** This 27-item version measures **psychopathy**, **narcissism** and **Machiavellianism**

using a 5-point Likert scale. We used this questionnaire, because the one we had used in Study 1 [54] showed weaker psychometric properties [44, 63].

**Big-Six questionnaire (29QB6) [45]** (The French translation was done through translation and back translation, the report on the formal validation is currently prepared for publication. In the meantime, the translation can be retrieved here [64]. The 29QB6 assesses personality along six dimensions: i) Extraversion, ii) Conscientiousness, iii) Honesty/Propriety, iv) Resiliency vs. Internalizing Negative Emotionality, v) Agreeableness, and vi) Originality/Talent. Higher **Extraversion** scores indicate that participants score higher on traits like talkativeness, sociability, assertiveness., gregariousness, and positive emotionality measures. **Conscientiousness** items refer to being organized, purposeful, and self-controlled. **Honesty/Propriety** items measure ethical behavior, integrity and deceit, instrumental use of others as well as aspects related to negative valence. **Resiliency** items capture the ability to internalize negative emotionality as a reverse of Neuroticism. Lower scores on resiliency resemble higher scores on neuroticism [45] **Agreeableness** items measure kindness and even temper. **Originality/Talent** items measure perceived talents and abilities, intellectual and aesthetic interest. They also include positive valence content. The 29 items are rated on a 6-point Likert scale, with 14 being reversely coded (see Table 2). Higher scores indicate higher level of expression of each personality trait (see Table 2).

**Trait Meta Mood Scale [65] (French version from Maria et al. [48]).** This 30-item questionnaire measures emotional intelligence by assessing the ability to monitor and manage one's own emotions. This scale consists in three subscales: i) **emotional attention**–the extent to which individuals attend to and value their feelings; ii) **emotional clarity**–the extent to which they feel clear about their feelings; and iii) **emotional repair**–the extent to which they use positive thinking to repair their negative mood. Participants rated each item on a 5-point Likert scale. Higher scores indicated higher levels of emotional attention, emotional clarity, and emotional repair (see Table 2).

**Balanced Inventory of Desirable Responding [66] (French version from D'Amours-Raymond [49]).** This 21-item questionnaire measures socially desirable responding along two dimensions: i) **self-deceptive positivity**–the tendency to give self-reports believed to have a positivity bias, and ii) **impression management**–deliberate effort at deception. We calculated the total score by summing all the responses on the 7-point Likert scale. We followed authors' recommendations and coded presence of social desirability (yes) when participants scored between 6 and 7, and absence of social desirability (no) for values below 5 (see Table 2).

**Other questionnaires.** The remaining self-reported questionnaires were the same as in Study 1: i) **General empathy score** using the 28-item Interpersonal Reactivity Index, ii) **Cultural dimensions** using the 26-item Individual Cultural Values Scale, and iii) **trust levels** using the World Value Survey 5. All our questionnaires demonstrated good or acceptable internal consistency (see Table 2).

**Self-reported lie detection abilities.** We asked participants to indicate if they were able to recognize somebody lying (yes/no answer) and rate their self-reported lie detection abilities on a continuous scale ranging from 0 (*unable to detect lies*) to 100 (*perfectly able to detect lies*). Afterwards, we asked participants to report their lie detection abilities in comparison to others from 0 (*much worse than others*) to 100 (*much better than others)* (see also [23–25, 51]).

## Procedure

The procedure was comparable to Study 1 in recruitment and programming, as well as in the sequence of events (see Fig 1). Participants completed the Cultural Values Scale, the Interpersonal Reactivity Index, their level of trust, the Dark Triad, the Balanced Inventory of Desirable

**Table 2. Description of the self-report questionnaires.**

| Construct and Scale | Subscales (items) | Response mode | Mean (SD) | Normative values* Mean (SD) | Cronbach's alpha |
|---|---|---|---|---|---|
| **Dark personality traits (Dark Triad)** *The French version of the Short Dark Triad* | psychopathy (9) narcissism (9) Machiavellianism (9) | 1 (*strongly disagree*) to 5 (*strongly agree*) | 2.11 (0.42) 2.63 (0.59) 2.85 (0.79) | 1.93 (1.01) 2.73 (0.75) 2.67 (0.99) | .727 .707 .672 |
| **Cultural dimensions** *Cultural Values Scale* | power distance (5) uncertainty avoidance (5) collectivism (6) masculinity (4) long-term orientation (6) | 1 (*strongly disagree*) to 7 (*strongly agree*) 1 (*extremely unimportant to me*) to 7 (*extremely important to me*) | 1.77 (0.27) 5.15 (0.29) 3.21 (0.47) 5.39 (0.71) 2.09 (0.31) | 2.15 (ns) 5.28 (ns) 3.70 (ns) 5.55 (ns) 2.83 (ns) | .723 .808 .817 .667 .718 |
| **Trust level** *The World Value Survey 5* | in group trust level (3) out group trust level (3) | 1 (*not at all*) to 4 (*completely*) | 3.04 (0.90) 2.58 (0.72) | 3.08 (ns) 1.72 (ns) | .456 .708 |
| **Empathy** *Interpersonal reactivity index* | **General Empathy score** by adding: empathic concern (7) perspective taking (7) fantasy (7) personal distress (7) | 0 (*does not describe me well*) to 4 (*describes me very well*) | 3.65 (0.84) 4.10 (0.78) 3.65 (0.79) 3.86 (1.16 2.96 (1.11) | 4.53 (1.06) 5.38 (0.88) 5.02 (0.96) 4.25 (1.26) 3.47 (1.13) | .786 .535 .500 .801 .800 |
| **Social Desirability** *The Balanced Inventory of Desirable Responding* | General score of social desirability (21) | 1 (*completely wrong*) to 7 (*completely true*) | .333 (0.14) | .364 (0.19) | .506 |
| **Emotional intelligence** *TMMS* | emotional attention (12) emotional clarity (10) emotional repair (6) | 0 (*strongly disagree*) to 6 (*strongly agree*) | 3.73 (0.40) 3.02 (0.38) 3.29 (0.37) | 3.32 (0.60) 3.63 (0.69) 3.59 (0.76) | |
| **Personality traits** *QB6* | Extraversion (5) Conscientiousness (5) Honesty/Propriety (5) Resiliency (4) Agreeableness (5) Originality/Talent(5) | 1 (*does not describe me well*) to 6(*describes me very well*) | 4.36 (0.90) 3.86 (1.04) 4.05 (0.89) 2.65 (0.38) 3.72 (1.01) 4.18 (0.71) | | .690 .764 .670 .805 .750 .607 |

*Note*. Description of the self-reported questionnaires, their subscales, and response mode. In addition, this table shows means, standard deviations (*SD*), normative values and Cronbach's alpha.

*References for the normative values: Dark Triad (N = 405 French-Canadian participants, M$_{age}$ = 31.01 years, SD$_{age}$ = 11.97, age range from 18 to76 years; [44]); cultural values (N = 223 French students, 104 males, age ranging from 18 to 35 years; [42]); trust level (N = 1089 adults from fifty societies [67]); empathy (N = 322 French participants, M$_{age}$ = 49.5 years, SD$_{age}$ = 21.1, age ranging from 18 to 89 years; [41]); social desirability (N = 1159 French-Canadian participants, 567 males, age ranging from 17 to 67 years [51]); emotional intelligence (N = 824 French undergraduate students, 368 males, M$_{age}$ = 20.7 years, SD = 2.1; [48]).

**Table 3. Results of the two multiple regression analysis on the continuous 22 predictor variables for self-reported lie detection abilities and self-reported lie detection abilities in comparison to others.**

| Predictors | Self-reported lie detection abilities | | | Self-reported lie detection abilities in comparison to others | | |
|---|---|---|---|---|---|---|
| | β (SD) | 95% CI | p value | β (SD) | 95% CI | p value |
| Machiavellianism | -0.22 (1.95) | [-4.06, 3.6] | .908 | .027 (1.50) | [-2.93, 3.0] | .986 |
| Psychopathy | 3.18 (2.05) | [-0.85, 7.2] | .122 | 3.62 (1.58) | [0.51, 6.7] | .023* |
| Narcissism | 1.97 (2.32) | [-2.58, 6.5] | .395 | 1.089 (1.79) | [-2.44, 4.6] | .546 |
| Power distance | -2.19 (1.42) | [-4.98, 0.6] | .123 | -.155 (1.10) | [-2.31, 2.0] | .888 |
| Uncertainty avoidance | -1.13 (1.18) | [-3.46, 1.2] | .338 | -.675 (.91) | [-2.47, 1.1] | .460 |
| Collectivism | -0.27 (0.89) | [-2.03, 1.5] | .761 | -1.13 (.69) | [-2.48, 0.2] | .104 |
| Masculinity | 0.34 (0.84) | [-1.31, 2.0] | .689 | -.886 (.65) | [-2.16, 0.4] | .173 |
| Long-term orientation | -0.99 (1.54) | [-4.02, 2.0] | .518 | -1.83 (1.19) | [-4.16, 0.5] | .125 |
| General Empathy | 0.35 (0.45) | [-0.54, 1.2] | .438 | .471 (.35) | [-0.22, 1.2] | .178 |
| In group trust level | -3.04 (2.12) | [-7.21, 1.1] | .152 | -.864 (1.64) | [-4.08, 2.4] | .598 |
| Out group trust level | -3.34 (1.50) | [-6.30, -0.4] | .026* | -1.42 (1.16) | [-3.70, 0.9] | .220 |
| Honesty/Propriety | -1.84 (1.20) | [-4.20, 0.5] | .126 | -1.52 (.93) | [-3.34, 0.3] | .103 |
| Resiliency | -0.19 (1.24) | [-2.63, 2.3] | .877 | .091 (.96) | [-1.80, 2.0] | .925 |
| Extraversion | -0.96 (1.23) | [-3.38, 1.5] | .437 | -1.02 (.95) | [-2.88, 0.9] | .285 |
| Agreeableness | -1.36 (1.00) | [-3.33, 0.6] | .177 | -0.95 (.78) | [-2.47, 0.6] | .223 |
| Conscientiousness | 1.04 (1.01) | [-0.94, 3.0] | .302 | .938 (.78) | [-0.59, 2.5] | .229 |
| Originality/Talent | 2.47 (1.57) | [-0.61, 5.5] | .116 | 3.98 (1.21) | [1.61, 6.4] | .001*** |
| Emotional repair | -0.542 (1.21) | [-2.91, 1.8] | .653 | .538 (.93) | [-1.29, 2.4] | .563 |
| Emotional attention | -1.22 (1.71) | [-4.58, 2.1] | .476 | -2.06 (1.32) | [-4.65, 0.5] | .119 |
| Emotional clarity | 0.51 (1.38) | [-2.21, 3.2] | .712 | -.486 (1.07) | [-2.59, 1.6] | .650 |
| Social desirability | 0.67 (0.33) | [0.01, 1.3] | .046* | .585 (.26) | [0.08, 1.1] | .023* |
| Gender (male) | -1.80 (2.52) | [-6.76, 3.2] | .475 | 2.31 (1.95) | [-1.51, 6.1] | .235 |

*Note.* The table displays standardized coefficients (β), standard errors, p-values associated with each predictor of the regression predicting the self-reported lie detection abilities and the self-reported lie detection abilities in comparison to others.

***$p < .001$

**$p < .01$

*$p < .05$

Responding, the Trait and Meta Mood Scale, and the Big-Six questionnaires (Fig 1C-Study 2). Afterwards, participants self-reported their lie detection abilities (Fig 1D-Study 2). The entire study took about 20 minutes to complete.

## Statistical analysis

We performed two multiple regression analyses to test whether our individual difference measures predicted i) participants' self-reported lie detection abilities score (0–100), and ii) participants' self-reported lie detection abilities in comparison to others (see Table 3). We present results of the correlations between all predictor variables in supporting information (see S3 Table).

## Results

Most participants (76.94%) reported being able to detect lies (yes/no answer). Many participants (62.69%) reported that their abilities to detect lies was above chance level (i.e., higher than 50% of accuracy). However, when asked to rate their abilities in comparison to others, only 20.46% of the participants reported being better than others at detecting lies. Most

participants (59.59%) estimated their abilities to be below those of others. However, a Pearson correlation on these two self-report measures was significant, $r(386) = .683$, $p < .001$ (see S3 Table), higher self-reported lie detection abilities correlated with higher self-reported lie detection abilities in comparison to others.

The overall multiple regression model on self-reported lie detection abilities was significant, $F(363, 22) = 2.49$, $p < .001$, $R_{adj}^2 = .078$. Results indicated that the model explained 7.8% of the variance (see Table 3). As shown in Table 3, lower out-group trust levels and higher social desirability predicted higher self-reported lie detection abilities. The remaining predictor variables were not significant (see Table 3).

The multiple linear regression analysis on self-reported lie detection abilities in comparison to others was significant, $F(363, 22) = 3.57$, $p < .001$, $R_{adj}^2 = .128$. The model explained 12.8% of the variance (see Table 3). Again, enhanced social desirability significantly predicted enhanced self-reported lie detection abilities in comparison to others (see Table 3). We also found that higher originality and psychopathy predicted higher self-reported lie detection abilities in comparison to others (see Table 3). The remaining predictor variables were not significant (see Table 3).

## Discussion

The literature on lie detection demonstrates a little understood paradox. Subjectively, many people self-report being above average in lie detection abilities [21, 23–25, 50], while objectively, at least on a group level, people perform around chance level [15, 16]. In two successive online studies, we investigated if and how individual differences can predict self-reported lie detection abilities. In Study 1, we measured Dark Triad personality traits, empathy, cultural values, and trust levels. In Study 2, we additionally accounted for conventional personality traits, emotional intelligence, and social desirability. As outcome measures, participants indicated whether they could detect lies (dichotomous yes/no answer), and to what extent. In Study 1, we coded extent as above versus below chance level, and in Study 2 as scores on a 100-point rating scale. In Study 2, we also asked participants to rate their self-reported lie detection abilities in comparison to others (see also [23–25, 51]).

In both studies, over 75% of our participants indicated being able to detect lies. When asked about extent, most participants rated their accuracy above chance level (i.e., being 50% of the time correct). Thus, we replicated that individuals on the group level consider themselves able to detect lies (e.g., [21, 23–25, 50]), which is well above the performance one could expect from actual lie detection tasks [15, 16]. When asked to rate their abilities in comparison to others, we found, however, that only about 20% of our participants indicated *yes*, they judged themselves to be superior to others at detecting lies. For extent, the numbers were comparatively low too; participants indicated being only 40% of the time better than others at detecting lies.

Our primary study goal was to investigate individual differences in self-reported lie detection abilities. Contrary to our expectations, we found few systematic relationships. In Study 1, the overall model included 12 predictor variables and was not significant. In Study 2, the overall model included 22 predictor variables and the model was significant. We observed enhanced self-reported lie detection abilities with i) decreasing out-group trust level scores, and ii) increasing social desirability scores. Neither gender, personality traits (i.e., Dark Triad, Big-Six personality traits), social cognition measures (i.e., empathy and emotional intelligence), nor cultural values predicted self-reported lie detection abilities. Interestingly, our individual difference measures were better predictors of self-reported lie detection abilities in comparison to others. Namely, enhanced psychopathy, enhanced Originality/Talent, and enhanced social desirability predicted enhanced self-reported lie detection abilities in

comparison to others. The findings on psychopathy and Originality/Talent (i.e., Openness) have been reported previously [22, 51].

Given our unexpected results, we decided to focus on three major observations. First, very few of our individual difference measures predicted self-reported lie detection abilities. We discuss in detail results on the Dark triad personality traits, because we had the strongest predictions for these traits [21–25]. Second, we replicated two previous results, the one on psychopathy [22] and the one Originality/Talent [51]. Yet, these replications were observed when looking at self-reported lie detection abilities in comparison to others. Third, we observed that increased social desirability scores linked to enhanced self-reported lie detection abilities in general as well as in comparison to others.

When designing the studies, we selected individual difference measures that have been mentioned in the lying literature before [20, 22, 23, 25, 33, 34, 37, 39]. Most obvious was the Dark Triad, because of what it stands for, malevolent behavioral tendencies such as self-promotion, manipulation (including lying), and lack of empathy [21–25]. Moreover, the Dark Triad had been previously linked to self-reported lie detection abilities [22–25]. We found that higher psychopathy scores were associated with higher self-reported lie detection abilities, but only when participants were asked to indicate their abilities in comparison to others. Wissing and Reinhard [22] found a similar relationship, but for higher confidence in self-reported lie detection performances. Thus, we found similar results despite studies using different response modes, suggesting that response mode does not matter. Against this suggestion, we did not replicate that higher levels of narcissism were associated with a higher self-reported lie detection abilities in comparison to others [23–25]. Likewise, we did not find that higher levels of narcissism were associated with higher self-reported lie detection abilities.

It is possible that differences between Dark Triad studies are due to differences in populations and measurements. For populations, this possibility is unlikely, because all relevant Dark Triad studies tested undergraduate university students from Western countries. In our case, we tested French speakers, largely in Switzerland, once 487 (99 males) participants and once 386 participants (72 males). Another study tested 207 participants (83 males) from Germany [22], and still other studies tested, respectively, 125 males [23], and 100 undergraduate students (15 males) [25] from Israel. These populations were comparable in gender composition and age, several being biased towards females and others towards males, with the mean population age not exceeding 30 years old.

For measurements, studies differed. We used different Dark Triad questionnaires in Study 1 and Study 2, both being shortened versions [40, 44, 54, 62]. In Study 1, we had used the 12-items Dirty Dozen questionnaire [40, 54], and did not replicate previous findings [22–25]. As questionnaire brevity comes with costs, such as reduced construct validity [63], we used the longer and psychometrically superior 27-items Short Dark Triad questionnaire in Study 2 [44, 62]. However, results were comparable for Study 1 and Study 2. One possible explanation could be questionnaire length, because relationships with narcissism scores were found for the 40-item Narcissistic Personality Inventory [23–25, 68, 69]. Yet, length cannot explain it all, because relationships between elevated psychopathy scores, overall Dark Triad scores, and confidence in lie detection abilities had been found using an even shorter 9-item scale [22, 70]. As of now, we have insufficient evidence to argue that divergent findings were due to populations or measurements.

Overall, we found hardly any individual difference measures, significantly predicting self-reported lie detection abilities. The situation was not really different when participants rated their lie detection abilities in comparison to others. We found that social desirability scores predicted enhanced ratings of both self-reported lie detection measures. Thus, self-reported lie detection abilities, whether in general or in comparison to others, tap into something that is

akin to an attitude or a social value. Maybe, this latter proposition might also explain why lower levels of out-group trust predicted higher self-reported lie detection abilities. In this regard, self-reported lie detection abilities might reflect people's *truth-default* state [20] which implies that we assume that others are trustworthy. In addition, indicating that one is able to detect lies reflects on people's favorable self-view [21, 71, 72]. In other words, it is better for one's self-esteem to think that one is not easily deceived.

## Study limitations, challenges and future directions

We worked on self-reported lie detection abilities without assessing people's actual lie detection performance. Notwithstanding, on a group level, we assume that our populations' actual lie detection performances would mirror results of previous studies showing that lie detection performance is around chance level [15, 16]. Future studies should assess both actual and self-reported lie detection abilities in a within-subjects design.

As with most studies in psychology, we tested undergraduate psychology students. They are not representative of the general population, whether within or between cultures (see e.g., [38, 73, 74]). Therefore, the generalizability of our results remains questionable. To encompass this limitation, we are currently running further studies testing alternative populations such as particular groups of professionals who have a relatively enhanced probability encountering deception (i.e., police officers, teacher, and insurers). We will report these results in the future. Moreover, further studies could consider age, because studies showed that lie detection abilities decreased with age [75–78] and lies were spotted more easily in older than younger people [78].

Worth noting, we did not replicate that our participants reported being above average in self-reported lie detection abilities, i.e., being in average better when comparing oneself to others (e.g., [21, 23–25, 50]). This observation reflects the *better-than-average effect* [21, 79]. On the contrary, only 20% of our participants thought they were better than others. When asked about extent, only 40% of them indicated being better than others at detecting lies. We could have expected this *better-than-average effect*, because it seems more pronounced for young people, and for dimensions that lack external verification [72]. These conditions both apply to our study. Another observation, we might have found comparable results with previous studies using self-reported lie detection abilities in comparison to others, if we would have highlighted the middle point of the rating scale, just like the other studied did. Previous studies indicated 50 as the point at which participants rated themselves as good as others [23–25]. In contrast, we asked participants to use the scale from 0 (*much worse than others*) to 100 (*much better than others*). Therefore, the average is not explicitly presented to the participants.

As a final critical point, some of the self-report questionnaires lacked validity [63, 80, 81]. Returning to the Dark Triad, these self-report questionnaires seemed to extract the uniqueness between narcissism, psychopathy, and Machiavellianism [82]. Several studies, however, reported an overlap between the three constructs and claimed they were not independent [81, 83]. Kajonius et al. [80] even argued that the Dirty Dozen questionnaire assesses only two constructs, namely, narcissism and an anti-social trait (i.e., Machiavellianism and psychopathy combined). Consistent with this, Persson et al. [84] showed that Machiavellianism and psychopathy, measured by the Short Dark Triad, were similar concepts. Therefore, future studies investigating similar questions to ours should include different measures, such as the Narcissistic Personality Inventory [69], because the latter resulted in comparable findings between studies [23–25]. Regarding social desirability, several studies emphasized the lack of validity of the BIDR (Balanced Inventory of Desirable Responding) [85, 86]. To address this limitation, further studies are needed to measure social desirability as a trait, using validated measures of related dimensions such as self-control (e.g. [87]).

## Conclusion

In two online studies, we tested self-reported lie detection abilities. On the group level, we replicated that people reported being above chance in lie detection. For individual difference measures, we observed that participants had higher lie detection abilities with increasing social desirability and decreasing out-group trust levels, respectively. Unlike our predictions, all other predictor variables were not significant, including the Dark triad and conventional personality traits, empathy, emotional intelligence, and cultural values. When participants rated their lie detection abilities in comparison to others, we did not find the *better-than-average effect*. Yet, we replicated that enhanced psychopathy and Originality/Talent predicted the latter measure. Our results indicated that it was socially desirable to trust others and to believe being able to detect lies. Future studies should test whether self-reported and actual lie detection abilities are associated and in which way, as well as assess these relationships in different professional groups, different age groups, and in populations beyond the frequently researched Western societies.

## Supporting information

**S1 Table. Results of the binary logistic regression analysis on the continuous 12 predictor variables for self-reported lie detection abilities (below vs above chance level).**
(TIF)

**S2 Table. Pearson's correlations between individual characteristics and self-reported lie detection abilities for Study 1.**
(TIF)

**S3 Table. Pearson's correlations between individual characteristics and self-reported lie detection abilities for Study 2.**
(TIF)

## Author Contributions

**Conceptualization:** Mélanie Fernandes, Christine Mohr.

**Data curation:** Mélanie Fernandes, Domicele Jonauskaite, Eric Laurent, Christine Mohr.

**Methodology:** Mélanie Fernandes, Eric Laurent, Christine Mohr.

**Supervision:** Christine Mohr.

**Writing – original draft:** Mélanie Fernandes, Domicele Jonauskaite, Frédéric Tomas, Christine Mohr.

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
