## [Decision Letter · Decision Letter 0]

21 Nov 2022

PONE-D-22-29409Individual differences in self-reported lie detection abilities.PLOS ONE

Dear Dr. Fernandes,

Thank you for submitting your manuscript to PLOS ONE. After careful consideration, we feel that it has merit but does not fully meet PLOS ONE’s publication criteria as it currently stands. Therefore, we invite you to submit a revised version of the manuscript that addresses the points raised during the review process. Please submit your revised manuscript by Jan 05 2023 11:59PM. If you will need more time than this to complete your revisions, please reply to this message or contact the journal office at plosone@plos.org. Please include the following items when submitting your revised manuscript:A rebuttal letter that responds to each point raised by the academic editor and reviewer(s). You should upload this letter as a separate file labeled 'Response to Reviewers'.A marked-up copy of your manuscript that highlights changes made to the original version. You should upload this as a separate file labeled 'Revised Manuscript with Track Changes'.An unmarked version of your revised paper without tracked changes. You should upload this as a separate file labeled 'Manuscript'.

We look forward to receiving your revised manuscript.

Kind regards,

Peter Karl Jonason

Academic Editor

PLOS ONE

Reviewers' comments:

Reviewer's Responses to Questions

**Comments to the Author**

1. Is the manuscript technically sound, and do the data support the conclusions?

Reviewer #1: Yes

Reviewer #2: Yes

Reviewer #3: Partly

2. Has the statistical analysis been performed appropriately and rigorously? 

Reviewer #1: No

Reviewer #2: Yes

Reviewer #3: Yes

3. Have the authors made all data underlying the findings in their manuscript fully available?

Reviewer #1: Yes

Reviewer #2: Yes

Reviewer #3: Yes

4. Is the manuscript presented in an intelligible fashion and written in standard English?

Reviewer #1: Yes

Reviewer #2: Yes

Reviewer #3: No

5. Review Comments to the Author

Reviewer #1: The present research used a correlational design to search for variables that may explain the gap between above-average self-assessed lie-detection ability and the average performance of lie detection. They used a large sample in two studies to again show that the bias exists and that people overestimate their lie-detecting ability. The perceived lie-detection ability was compared with different self-report scales, and in most cases, negative results were obtained. The outcomes are essential to enrich our knowledge and guide the exploration of the bias in additional directions. Furthermore, the perceived lie-detection ability is vital in face-to-face communication and may guide behavior. Undoubtedly, the topic deserves more empirical attention, and any addition to our limited knowledge in this domain is welcome.

Nevertheless, I have some comments that can and should be addressed in a revision.

Any ability is defined on a continuous scale. The dichotomous yes/no response is meaningless and redundant. People may succeed more or less in lie detection. No one is perfect, and no one is 100% inaccurate. Further, the dichotomization triggers odd results, such as 9 participants who answered that they were not able to detect lies and at the same time indicated that they succeeded in about 50-75% of their lie detection attempts. Furthermore, 4 participants who answered yes to the yes/no lie detection ability question indicated success in only 0-25% of their attempts (Table 2). The total frequency line (bottom) demonstrates the overestimated lie-detection bias, and the dichotomization adds nothing in this respect.

The 0 to 100% scale used in Study 2 is more sensitive than the scale used in Study 1. Therefore, why not benefit from the advantage and use parametric statistics? Instead, the authors dichotomized the scale to use a non-parametric chi-square analysis (line 362, on). Note that the sample consisted of 386 participants, which calls for parametric statistics. Further, the contradiction between answering no to the absolute lie detection ability and receiving an above-chance level score when reporting the lie detection ability persists. Therefore, the manipulation check in Table 5 is meaningless. In addition, only the bottom line in Table 6, which shows the lie-detection bias, is relevant. In sum, I suggest removing the absolute yes/no lie-detection ability test.

Line 476: In study 2, we also asked them (the participants) to report their ability in comparison to others. Unfortunately, the results were not reported here. The authors indicated that these results would be reported elsewhere. My question is, why? This is an essential addition to the present study, and the comparison between answering with and without reference to others is interesting and important (to my knowledge, the comparison to other people results in lower absolute scores).

Minor points:

Line 55: The current view about frequent lying is that not many people lie frequently, and most people reported not lying in the previous 24 h. (Daiku et al. 2021; Halevi et al. 2014; Serota and Levine 2015).

Daiku, Y., Serota, K. B., Levine, T. R. (2021). A few prolific liars in Japan:

Replication and the effects of Dark Triad personality traits. PLoS ONE 16(4):

e0249815. https://doi.org/10.1371/journal.pone.0249815

Halevy, R., Shalvi, S., and Verschuere, B. (2014). Being honest about dishonesty:

Correlating self-reports and actual lying. Human Communication Research, 40,

54-72.

Serota, K. B., & Levine, T. R. (2015). A few prolific liars: Variation in the prevalence of lying. Journal of Language and

Social Psychology, 34(2), 138-157.

Line 91: … self-reported confidence in own lie detection abilities was unrelated to individuals' actual lie detection abilities… add a reference (for example, Elaad and Gonen-Gal, 2022).

Elaad, E. & Gonen-Gal, Y. (2022). Face-to-face lying: Effects of gender and motivation

To deceive. Frontiers in Psychology: Forensic and Legal Psychology, 13: Article

820923. doi: 10.3389/fpsyg.2022.820923.

Line 214: Relative self-reported lie detection ability. Relative to what? To others? to the average person? The authors relate to the extent or degree of the perceived lie detection ability not to relativeness. This should be corrected.

Lines 230-231, 297: How were the questionnaires completed? Individually or in groups? If in groups, how big were the groups? Please be specific.

Line 236: Separate detect and changes.

Line 282: Replace Is with It.

Line 361: Why is the power set at 0.80?

Line 371: Change relative to relatively sensitive (The 0 to 100 scale is more sensitive than the categorial scale used in study 1).

Line 377: The scale ranges from 0 to 100. Remove the percent sign (100%).

Line 488: Elaad and Reizer (2015) used the Big Five, which is a different scale than the Big Six that was used in the present study.

Line 424: Not adding actual lie-detection performance is indeed a limitation. The current trend is to compare lie-detection ability scale scores with actual behavior (See Elaad and Gonen-Gal, 2022).

Reviewer #2: This paper examined individual differences (primarily of personality) on self-reported lie detection ability. The Introduction makes a solid argument for the need for the study. Individual difference mostly had few significant or strong relationships with self-reported lie detection ability. This is interesting as some have argued that they should. I do not consider the lack of strong or significant effects to be a problem, they are what they are.

The paper reports 2 studies that appear to have been properly executed with samples of a good size and well-selected measures. Data are reported in a way that address the questions of the studies, although I’d like to see tables showing the intercorrelation of all predictor variables in the supplementary material to aid in the interpretation of the regressions.

The introduction is clear and readable. I have no suggested changes

Study 1

Method

Participant numbers don’t add up or there may be an error. It is stated in the Participants section that 525 were recruited. In the Data preparation section, it is stated that 239 were excluded, leaving 487 = these figures don’t match.

Additionally, it is stated in the Participants that there were 487 participants (95 men) from one university, and this is given as the final sample size in the Data preparation section – is this a coincidence or an error in copying numbers?

Results

I can’t find how gender was coded, thus the mean and any direction of relationships cannot be interpreted by a reader.

Study 2

Method

The participants (700) minus exclusions (234) does not add to the total final sample (386). There was an age exclusion but the numbers are not stated.

Results

Why is gender omitted in the Study 2 regression analysis?

Discussion

One issue to be aware of it that a recent meta-analysis of social desirability scales suggests they are of no value (Lanz et al, 2022). Although social desirability is one of only 2 significant predictors in Study 2, it is worth asking the extent to which this is truly meaningful given the new analysis of the relevant measures.

Lanz, L., Thielmann, I., & Gerpott, F. H. (2022). Are social desirability scales desirable? A meta‐analytic test of the validity of social desirability scales in the context of prosocial behavior. Journal of Personality, 90(2), 203-221.

Reviewer #3: Verse 122. This research information should be in the method part, not in the theoretical introduction.

Verse 150 . Why is the sample of respondents (although numerous) exclusively students? After all, it is known that this is a very specific group when it comes to self -report research .

From verse 157 . Why such a long description of individual scales and tools when you can include all the most important information in a table. A lot of information is duplicated.

Table 1 : Cronbach 's alpha on the " psychopathy " scale is too low for analysis. Similarly, too low " Cronbach 's alpha " is on the in group trust level scale .

From verse 218 . Why is there so much information about the procedure and participants in the text itself? There is a diagram illustrating the procedure at the end, and the necessary information can be included in a table. This takes up a lot of space.

From verse 243 . Maybe I'm repeating myself, but why such a long description again when it can be included in a few sentences or a clear table?

Table 3 Should scales with such low reliability be included in the analysis ( Psychopathy and in - group trust)?

From verse 292. Another test of students, and in psychology at that. Is this group representative in the self report?

From verse 299 . Again, why are they introducing so many new research tools plus including the ones they used in the previous study. Do they want to measure everything in this article?

Table 4 . Again, we have quite poor reliability on several scales: in trust group , empathic concerns , perspective taking , general score of social desirability .

From verse 338 . Do you really need to re-detail what is contained in the table?

Verse 406 . Isn't 21 self -report variables too much for one article?

Verse 415 . They put in so many variables and only 13 percent of the variance is explained? It's probably a very small number. Maybe a bad theory?

Verse 442 . Of the 21 variables, only 2 are significant predictors .

Verse 447 . Since such a small number of significant variables was unexpected, perhaps it was better to start the theories earlier instead of coming to such conclusions only now after a huge amount of work.

Verse 483-485 . I don't understand this line of reasoning. On what theoretical basis is such a conclusion?

Verse 529 . Instead of a larger sample of respondents, maybe it's better not to study only psychology students?

Overall, there is a lot of chaos in this article and it lacks a cohesive structure. The authors tackle an interesting problem, but they are actually lost in the large amount of research and variables introduced that don't really explain the phenomenon of detecting lies in other people. After reading this article, it is basically unknown what these individual differences in the title would consist of.

The article is too long-winded. It seems like it could be half as long. It contains numerous repetitions and doubles. In the theoretical part, the authors briefly cite a lot of variables and studies related to lie detection, the description of which, instead of explaining this phenomenon, only complicates its understanding. In addition, in the theoretical part, they mix theory with the method that should be in the next part. Again, there is the problem of text systematization and structure

In the methodological part, the selection of samples is puzzling. Maybe students (including psychology) are not necessarily a representative group for such a common phenomenon as lying and its detection. Then there is the issue of a large number of questionnaires and variables for one article.

The article contains numerous mental abbreviations that are not always understandable to the reader. The structure of sentences, which is sometimes too complicated, does not help to understand the article.

In conclusion, it seems that the authors put a lot of work into this article, but it is too vague for the reader.

The manuscript deals with an interesting topic, but in the manuscript too often the names can refer to "lie detection" which the authors have not studied. "lie detection" is not possible to examine by self-report. It can be assumed that when the authors wrote about "lie detection", they did not mean it specifically. It should be made clear in every part of the manuscript that the theory and the study itself are not about "lie detection", because using the methods presented in the manuscript variable cannot be evaluated. It will also be advisable to reduce the number of variables to only those that have theoretical justification. The revised text will certainly be very interesting.

6. PLOS authors have the option to publish the peer review history of their article (what does this mean?). If published, this will include your full peer review and any attached files.

Reviewer #1: No

Reviewer #2: **Yes: **Guy J. Curtis

Reviewer #3: No

---

## [Author Response · Author response to Decision Letter 0]

29 Mar 2023

Dear Editor, and editorial team, Dear Dr. Jonason, 

We are delighted by your invitation to revise and resubmit our manuscript entitled “Individual differences in self-reported lie detection abilities” to PLoS ONE. We would like to thank you and the three reviewers for the detailed comments. We believe that these have greatly helped us strengthen the quality of our manuscript. 

Please, find below our specific responses to the reviewers’ comments in bold. Changes in the text of the revised manuscript are highlighted in italic. We have incorporated almost all suggestions. In case we decided against implementation, we explain why in this response letter.

We hope that these changes to the manuscript will make it suitable for publication in PLoS ONE and we look forward to hearing from you at your earliest convenience.

Best regards, 

Mélanie Fernandes*, Domicele Jonauskaite, Frédéric Tomas, Eric Laurent, and Christine Mohr

*corresponding author

 

Reviewer #1: 

The present research used a correlational design to search for variables that may explain the gap between above-average self-assessed lie-detection ability and the average performance of lie detection. They used a large sample in two studies to again show that the bias exists and that people overestimate their lie-detecting ability. The perceived lie-detection ability was compared with different self-report scales, and in most cases, negative results were obtained. The outcomes are essential to enrich our knowledge and guide the exploration of the bias in additional directions. Furthermore, the perceived lie-detection ability is vital in face-to-face communication and may guide behavior. Undoubtedly, the topic deserves more empirical attention, and any addition to our limited knowledge in this domain is welcome.

Nevertheless, I have some comments that can and should be addressed in a revision.

R1.1. Any ability is defined on a continuous scale. The dichotomous yes/no response is meaningless and redundant. People may succeed more or less in lie detection. No one is perfect, and no one is 100% inaccurate. Further, the dichotomization triggers odd results, such as 9 participants who answered that they were not able to detect lies and at the same time indicated that they succeeded in about 50-75% of their lie detection attempts. Furthermore, 4 participants who answered yes to the yes/no lie detection ability question indicated success in only 0-25% of their attempts (Table 2). The total frequency line (bottom) demonstrates the overestimated lie-detection bias, and the dichotomization adds nothing in this respect.

Reply: The reviewer is right, the analysis of the dichotomous yes/no scale does not add any real value to the study but might create confusion. Therefore, in the revised manuscript, we merely present descriptive statistics on the yes/no answers. On page 13, we now write: 

“Overall, most participants (81.52%) reported being able to detect lies (yes/no answer) and many participants (55.44%) self-reported that their lie detection abilities ranged between 50% and 100% (i.e. above chance level self-reported lie detection abilities).”

Then, we also removed the Chi-Square tests of independence and the binary logistic regression analysis with 12 continuous predictor variables and yes/no answers as an outcome variable. We now computed a binary logistic regression analysis with these same 12 predictor variables and the self-reported lie detection ability score as an outcome variable (see page 12 of the revised manuscript):

“To test our study question, we conducted a binary logistic regression analysis on this dependent dichotomous variable (below vs. above chance level) using the following 12 continuous predictor variables: psychopathy, narcissism, Machiavellianism, power distance, uncertainty avoidance, collectivism, masculinity, long-term orientation, empathy, ingroup trust level, outgroup trust level, and gender (see also S1 Table). We presented the correlations between all the predictor variables as supporting information (see S2 Table).”

R1.2. The 0 to 100% scale used in Study 2 is more sensitive than the scale used in Study 1. Therefore, why not benefit from the advantage and use parametric statistics? Instead, the authors dichotomized the scale to use a non-parametric chi-square analysis (line 362, on). Note that the sample consisted of 386 participants, which calls for parametric statistics. Further, the contradiction between answering no to the absolute lie detection ability and receiving an above-chance level score when reporting the lie detection ability persists. Therefore, the manipulation check in Table 5 is meaningless. In addition, only the bottom line in Table 6, which shows the lie-detection bias, is relevant. In sum, I suggest removing the absolute yes/no lie-detection ability test.

Reply: We deleted the analyses on the yes/no scale from both Studies 1 and 2 (see also our reply to the previous comment R1.1). We followed this reviewer’s suggestion to work with the continuous self-reported lie detection score and adopted parametric statistics. In the revised version of the manuscript, on page 18, we changed the statistical analysis section: 

“We performed two multiple regression analyses to test whether our individual differences measures predicted i) participants’ self-reported lie detection abilities score (0-100), and ii) participants’ self-reported lie detection abilities in comparison to others (see Table 3). We present results of the correlations between all predictor variables in supporting information (see S3 Table).”

R1.3. Line 476: In study 2, we also asked them (the participants) to report their ability in comparison to others. Unfortunately, the results were not reported here. The authors indicated that these results would be reported elsewhere. My question is, why? This is an essential addition to the present study, and the comparison between answering with and without reference to others is interesting and important (to my knowledge, the comparison to other people results in lower absolute scores).

Reply: Following this comment, we reconsidered. We were interested in individuals’ belief biases, thus, what individuals think about their own lie detection abilities. Subsequent to the results of Study 1, we decided to additionally ask about individuals’ biases when compared to others, because this had been done in previous studies (Elaad, 2009; Zvi & Elaad, 2018; Elaad & Reizer, 2015, Elaad et al., 2020; Elaad, 2022). But we agree, we should give this measure more space in our analysis of Study 2. This decision had implications well beyond the method and result section, being also more extensively considered in the discussion.

In the statistical analysis section and in the result section (see page 18), we now respectively write: 

“We performed two multiple regression analyses to test whether our individual differences measures predicted i) participants’ self-reported lie detection abilities score (0-100), and ii) participants’ self-reported lie detection abilities in comparison to others (see Table 3). We present results of the correlations between all predictor variables in supporting information (see S3 Table).”

[…] “However, when asked to rate their abilities in comparison to others, only 20.46% of the participants reported being better than others at detecting lies. Most participants (59.59%) estimated their abilities to be below those of others. However, a Pearson correlation on these two self-report measures was significant, r(386) = .683, p < .001 (see S3 Table), higher self-reported lie detection abilities correlated with higher self-reported lie detection abilities in comparison to others.”

We also added on page 20 (results section):

“The multiple linear regression analysis on self-reported lie detection abilities in comparison to others was significant, F(363, 22) = 3.57, p <.001, Radj2 = .128. The model explained 12.8 % of the variance (see Table 3). Again, enhanced social desirability significantly predicted enhanced self-reported lie detection abilities in comparison to others (see Table3). We also found that higher originality and psychopathy predicted higher self-reported lie detection abilities in comparison to others (see Table 3). The remaining predictor variables were not significant (see Table 3).”

On page 22 (discussion section), we now write: 

” When asked to rate their abilities in comparison to others, we found, however, that only about 20% of our participants indicated that yes, they judged themselves superior to others at detecting lies. For extent, the numbers were comparatively low too; participants indicated being only 40 % of the time better than others at detecting lies.”

And also added on page 22 (discussion section):

“Interestingly, our individual difference measures were better predictors of self-reported lie detection abilities in comparison to others. Namely, enhanced psychopathy, enhanced Originality/Talent, and enhanced social desirability predicted enhanced self-reported lie detection abilities in comparison to others. The findings on psychopathy and Originality/Talent (i.e., Openness) have been reported previously [22, 51].”

Reference [22, 51] of the revised manuscript are:

22. Wissing BG, Reinhard MA. The Dark Triad and the PID-5 Maladaptive Personality Traits: Accuracy, confidence and response bias in judgments of veracity. Front Psychol. 2017 Sep 21;8:1549. doi: 10.3389/fpsyg.2017.01549.

51. Elaad E, Reizer A. Personality correlates of the self-assessed abilities to tell and detect lies, tell truths, and believe others. J Individ Differ. 2015;36(3):163-69. doi: 10.1027/1614-0001/a000168.

Minor points:

R1.4.Line 55: The current view about frequent lying is that not many people lie frequently, and most people reported not lying in the previous 24 h. (Daiku et al. 2021; Halevi et al. 2014; Serota and Levine 2015).

Daiku, Y., Serota, K. B., Levine, T. R. (2021). A few prolific liars in Japan:

Replication and the effects of Dark Triad personality traits. PLoS ONE 16(4):

e0249815. https://doi.org/10.1371/journal.pone.0249815

Halevy, R., Shalvi, S., and Verschuere, B. (2014). Being honest about dishonesty:

Correlating self-reports and actual lying. Human Communication Research, 40,

54-72.

Serota, K. B., & Levine, T. R. (2015). A few prolific liars: Variation in the prevalence of lying. Journal of Language and

Social Psychology, 34(2), 138-157.

Reply: We appreciate these literature suggestions and added the references in the revised version of the manuscript as proposed, see page 3:

“Yet, not everybody lies to the same extent with more recent studies showing that a few prolific liars are telling the majority of lies [6, 10, 11].”

References [6,10,11] of the revised manuscript corresponding to:

6. Serota KB, Levine TR. A few prolific liars: variation in the prevalence of lying. J Lang Soc Psychol. 2014 Apr 04;34(2):138-57. doi: 10.1177/0261927X14528804.

10. Daiku Y, Serota KB, Levine TR. A few prolific liars in Japan: replication and the effects of Dark Triad personality traits. PLoS One. 2021 Apr 15;16(4):e0249815. doi: 10.1371/journal.pone.0249815.

11. Halevy R, Shalvi S, Verschuere B. Being honest about dishonesty: correlating self-reports and actual lying. Hum Commun Res. 2014 Jan 01;40(1):54-72. doi: 10.1111/hcre.12019.

R1.5.Line 91: … self-reported confidence in own lie detection abilities was unrelated to individuals' actual lie detection abilities… add a reference (for example, Elaad and Gonen-Gal, 2022).

Elaad, E. & Gonen-Gal, Y. (2022). Face-to-face lying: Effects of gender and motivation to deceive. Frontiers in Psychology: Forensic and Legal Psychology, 13: Article820923. doi: 10.3389/fpsyg.2022.820923.

Reply: Idem to R1.4. We added the references in the revised version of the manuscript, see page 5: 

“Interestingly, self-reported confidence in one’s own lie detection abilities was unrelated to individuals’ actual lie detection abilities [26]. Also, variance in Dark Triad traits did not explain variance in actual lie detection abilities either [26].” 

Reference [26] is the suggested publication:

26. Elaad E, Gonen-Gal Y. Face-to-Face Lying: Gender and Motivation to Deceive. Front Psychol. 2022;13:820923. Epub 2022/04/09. doi: 10.3389/fpsyg.2022.820923. PubMed PMID: 35391990; PubMed Central PMCID: PMCPMC8982912.

R1.6.Line 214: Relative self-reported lie detection ability. Relative to what? To others? to the average person? The authors relate to the extent or degree of the perceived lie detection ability not to relativeness. This should be corrected.

Reply: We determined self-reported lie detection abilities in two ways: the absolute (yes/no) measure and the relative (0 to 100 scale) measure. We agree, the use of the term “relative” was not ideal in the earlier version of the manuscript. Having followed this reviewer’s suggestion (R1.1), we removed the absolute dichotomous variable (see our reply to R1.1). Therefore, the term “relative” became obsolete, we do not use it anymore in the revised version of the manuscript. 

R1.7.Lines 230-231, 297: How were the questionnaires completed? Individually or in groups? If in groups, how big were the groups? Please be specific.

Reply: The questionnaires were completed individually, which we explain on page 11 of the revised manuscript: 

“We used the LimeSurvey platform to prepare and run our online survey. We distributed the online link to potential volunteers. In case of interest, they could complete the survey at their own convenience. On the first two pages, we provided, respectively, written study information and ethical information, such as the right to withdraw from the study at any time, and data confidentiality (Fig 1A). We also stated that we treat participants’ continuation as informed consent. Next, participants provided socio-demographic information regarding their age, gender, nationality, and field of studies (Fig 1B). Then, they completed the self-reported questionnaires in the following order: The FR-C Dark Triad Dirty Dozen, the Cultural Values Scale, the IRI, and their level of trust based on the World Value Survey 5 (Fig 1C-Study 1 and Table 1). Afterwards, they indicated their self-reported lie detection abilities (Fig 1D-Study 1). Finally, they were fully debriefed and thanked for their participation. The survey took about 20 minutes to complete.“

R1.8. Line 236: Separate detect and changes.

Reply: We corrected this typo on page 12 of the revised manuscript: 

“The flow diagram depicts parts of the survey that i) were comparable to Study 1 and Study 2 (A, B), ii) were complemented by other questionnaires in Study 2 as compared to Study 1 (C), and iii) used different rating scales for self-reported lie detection abilities (D).”

R.1.9. Line 282: Replace Is with It.

Reply: This section has been removed (see response to comment R1.1). 

R1.10. Line 361: Why is the power set at 0.80?

Reply: The statistical power of a significant test is the probability of rejecting the null hypothesis, that is, avoiding a Type II error. According to Cohen (1988), setting the power at .80 implies a good balance between limiting the risk of Type II error and making the experiment feasible in terms of the sample size. A value below .80 would imply a great risk of failing to reject the null hypothesis (i.e., Type II error), and setting the value above .80 would demand larger sample sizes, which risk being difficult to recruit. Beyond this fixed value, we could also remind us that power increases with an increase in sample size. We tested 386 participants, 155 more than the minimum recommended number of 231 participants. We mention the sample size calculation on page 13 of the revised manuscript: 

“We used the statistical power analysis tool G*Power [52] to estimate our sample size. We determined a minimum sample size of 231 for a linear multiple regression test with a small effect size of 0.1, 21 predictors, α of 0.05, and power (1−β) of 0.80.”

Reference [52] of the revised manuscript corresponding to :

52. Faul F, Erdfelder E, Buchner A, Lang A-G. Statistical power analyses using G*Power 3.1: Tests for correlation and regression analyses. Behav Res Methods. 2009 Nov;41(4):1149-60. doi: 10.3758/BRM.41.4.1149.

R1.11. Line 371: Change relative to relatively sensitive (The 0 to 100 scale is more sensitive than the categorial scale used in study 1).

Reply: The sentence in question has been changed, because we deleted the distinction between “absolute” and “relative” abilities (see our response R1.6). 

R1.12. Line 377: The scale ranges from 0 to 100. Remove the percent sign (100%).

Reply: We removed the percent sign. 

R1.13. Line 488: Elaad and Reizer (2015) used the Big Five, which is a different scale than the Big Six that was used in the present study.

Reply: The reviewer is correct. Questionnaire selection is a challenge in a multi-lingual country (Switzerland). We were interested in using a questionnaire which would be suitable for cross-cultural research, not only because of the four national languages of Switzerland (German, French, Italian, and Romanch), but also to easily implement it in the cross-cultural study we are currently preparing. Due to this situation and vision, we chose the Big Six questionnaire, because it has been validated in 26 nations accounting for cultural differences in how people talk and describe personality (see Thalmayer & Saucier, 2014; Saucier, 2009). We also had translations to French readily available. We would like to add that much of the content of the Big-Five factors is covered by the Big Six factors. The Bix Six questionnaire additionally measure the factor for Honesty/Humility, which has been shown to be important outside the Anglo-Saxon populations (Thalmayer & Saucier, 2014). Big Six also has the factor named Resiliency, which should be interpreted as the reverse scale of Neuroticism (Saucier, 2009). Thus, studies are relevant to us whether using the Big-Five or the Big Six questionnaire. 

R1.14. Line 424: Not adding actual lie-detection performance is indeed a limitation. The current trend is to compare lie-detection ability scale scores with actual behavior (See Elaad and Gonen-Gal, 2022).

Reply: We totally agree with the reviewer that the study would have been stronger if we had measured actual lie-detection abilities. We had planned and prepared to do so, but COVID hit our laboratory study right at the beginning of data collection. Thus, as many others, we continued to run our studies online. Due to the local circumstances, we first worked with questionnaires, while also preparing means to test for actual lie detection performance. While running the here presented studies, we also created video material on lie detection in French. Thus, in parallel, we collected data on self-reported lie detection abilities and lie detection performance in students, comparing them to data gathered from various professional groups (e.g., policemen, teachers). Currently, we are treating these data to present them in a subsequent manuscript. Thus, the current manuscript represents the first empirical work realized by Melanie Fernandes for her doctoral thesis on self-reported lie detection abilities. In the current manuscript, we wanted to learn about individual differences measures that might be relevant for future studies we hoped to run under more controlled laboratory conditions. And indeed, we are currently preparing two more manuscripts in which we report on self-reported lie detection abilities and cues to lie detection as well as performances. We thank this reviewer for the recent (2022) reference, which will help us in our reasoning for all our ongoing studies. For the current manuscript, we also included this citation in our introduction, see response to comment R1.5.

Reviewer #2: 

This paper examined individual differences (primarily of personality) on self-reported lie detection ability. The Introduction makes a solid argument for the need for the study. Individual difference mostly had few significant or strong relationships with self-reported lie detection ability. This is interesting as some have argued that they should. I do not consider the lack of strong or significant effects to be a problem, they are what they are.

R2.1.The paper reports 2 studies that appear to have been properly executed with samples of a good size and well-selected measures. Data are reported in a way that address the questions of the studies, although I’d like to see tables showing the intercorrelation of all predictor variables in the supplementary material to aid in the interpretation of the regressions.

Reply: We thank Dr. Curtis for the positive feedback. We agree that it makes sense to report the intercorrelations of all predictor variables. They can be found in the supporting information section in S2 Table and S3 Table. 

R2.2. The introduction is clear and readable. I have no suggested changes

Reply: We thank Dr. Curtis for this positive feedback. 

R2.3.Study 1

Method

Participant numbers don’t add up or there may be an error. It is stated in the Participants section that 525 were recruited. In the Data preparation section, it is stated that 239 were excluded, leaving 487 = these figures don’t match.

Additionally, it is stated in the Participants that there were 487 participants (95 men) from one university, and this is given as the final sample size in the Data preparation section – is this a coincidence or an error in copying numbers?

Reply: This is an important observation for which we are grateful. We realized that the participant section needs further work to improve coherence and clarity. Accordingly, we now write on page 7 of the revised manuscript: 

“We recruited 764 participants (105 males). After excluding incomplete data and selecting participants between 18 and 31 years old (i.e., the majority), our final sample consisted in 487 participants (99 males) with a mean age of 21.50 years (SDage = 6.57 years; range = 18-31 years). Of these, 418 (95 males) were undergraduate students at the University of Lausanne, Switzerland, who received course credit for their participation. The remaining 69 participants (4 males) were recruited at the University of Franche-Comté, France. All participants were native French speakers.” 

R2.4.Results

I can’t find how gender was coded, thus the mean and any direction of relationships cannot be interpreted by a reader.

Reply: In the revised version, these results have been deleted. Yet, Dr. Curtis (see R2.6 below) also proposed a gender comparison, which we added to the binary logistic regression analysis (see S1 Table, on page 34 of the revised manuscript). Accordingly, we adapted the design and statistical analysis section (see page 12): 

“To test our study question, we conducted a binary logistic regression analysis on this dependent dichotomous variable (below vs. above chance level) using the following 12 continuous predictor variables: psychopathy, narcissism, Machiavellianism, power distance, uncertainty avoidance, collectivism, masculinity, long-term orientation, empathy, ingroup trust level, outgroup trust level, and gender (see also S1Table). We presented the correlations between all the predictor variables as supporting information (see S2 Table).”

R2.5.Study 2

Method

The participants (700) minus exclusions (234) does not add to the total final sample (386). There was an age exclusion, but the numbers are not stated.

Reply: We added the relevant information to this section (see revised manuscript, page 13) 

“We recruited a new sample of French speaking undergraduate psychology students (N = 700, 90 males) at the University of Lausanne. After excluding incomplete data and matching participants’ age to those of Study 1, we were left with 386 participants (72 males; Mage = 20.21, SDage = 2.22, range = 18 to 31 years). One academic year separated the data collection for Study 1 and Study 2. All participants received course credit for their participation.”

R2.6.Results

Why is gender omitted in the Study 2 regression analysis?

Reply: We have no strong reason to assume gender differences. Yet, we agree, we have no strong reason to assume that there are none. Accordingly, we now included gender and updated the results section of the revised manuscript accordingly (see Table 3 pages 19-20).

R2.7.Discussion

One issue to be aware of it that a recent meta-analysis of social desirability scales suggests they are of no value (Lanz et al, 2022). Although social desirability is one of only 2 significant predictors in Study 2, it is worth asking the extent to which this is truly meaningful given the new analysis of the relevant measures.

Lanz, L., Thielmann, I., & Gerpott, F. H. (2022). Are social desirability scales desirable? A meta‐analytic test of the validity of social desirability scales in the context of prosocial behavior. Journal of Personality, 90(2), 203-221.

Reply: We thank Dr. Curtis for informing us that our measure of social desirability (i.e. The Balenced Inventory of Desirable responding (BIDR), Paulhus, 1991) has limitations including validity. It seems that this scale neither measures socially desirable traits nor response biases (Holden & Passey, 2010; Lanz et al., 2021). We included this information on page 26: 

” Regarding social desirability, several studies emphasized the lack of validity of the BIDR (Balenced Inventory of Desirable Responding) [85, 86]. To address this limitation, further studies are needed to measure social desirability as a trait, using validated measures of related dimensions such as self-control (e.g.[87])”

References [85, 86] and [87] of the revised manuscript were:

85. Holden RR, Passey J. Socially desirable responding in personality assessment: Not necessarily faking and not necessarily substance. Pers Individ Dif. 2010 Oct;49(5):446-50. doi: 10.1016/j.paid.2010.04.015.

86. Lanz L, Thielmann I, Gerpott FH. Are social desirability scales desirable? A meta-analytic test of the validity of social desirability scales in the context of prosocial behavior. J Pers. 2022 Apr;90(2):203-21. doi: 10.1111/jopy.12662.

87. Tangney JP, Baumeister RF, Boone AL. High self-control predicts good adjustment, less pathology, better grades, and interpersonal success. J Pers. 2004 Apr;72(2):271-324. doi: 10.1111/j.0022-3506.2004.00263.x.

Reviewer #3: 

R3.1. Verse 122. This research information should be in the method part, not in the theoretical introduction.

Reply: We are not totally sure what exactly the reviewer wants us to do, because Verse 122 “… explain variance in self-reported lie detection abilities. In study 1, participants completed self-reported …”relate to a particular line or the paragraph more widely. This verse and the following lines explain details of our study, and some actual results from Study 1. When it comes to the details of Study 1 (method), we are not too sure what the reviewer wishes us to do, because we are used to quickly explain what we have done in any given study at the end of an introduction. We need to do so in order to formulate our hypotheses. If we do not convey what we have done, we cannot formulate what we expect the study to show. When it comes to the first results of Study 1, we felt we need to report this finding here in order to explain Study 2, because the outcome of Study 1 had informed Study 2. If there is anything we did not capture by reading the current comment, we would appreciate if they could be more specific about what should go to the method section.

R3.2. Verse 150. Why is the sample of respondents (although numerous) exclusively students? After all, it is known that this is a very specific group when it comes to self-report research.

Reply: The reviewer is absolutely right that studies in psychology should not exclusively focus on student populations. In 2010, Heinrich and colleagues introduced the term WEIRD populations, describing their criticism of testing predominantly students in Western, Educated, Industrialised, Rich and Democratic societies. By December 2022, their article has been cited 2,624 times. Now, does this study mean we should stop testing WEIRD populations? We are not sure who would and should have the final word to decide. What is certain, and very much in line with this referee’s comment, it should not be “exclusively” students as the world is more varied than that. A sample of students is a specific sub-group of the general population, and by inference not (necessarily) representative of the general population. It is for that very reason we run additional studies testing other populations. Currently, we are starting to treat data from different professional groups (police, education, insurance), couples, and prepare a cross-cultural study. The current study had been launched before the Covid pandemic and the lockdown hit countries worldwide. We had already prepared a follow-up study in the laboratory to also test our current participants’ actual lying detection ability (see also our reply to R1.14). Yet, we had to adapt to the sanitary situation, and so we had to drop the second part. In any case, we are now advancing in the treatment of the data from professional groups, and we hope to soon report these results in subsequent peer-reviewed contributions. 

R3.3. From verse 157. Why such a long description of individual scales and tools when you can include all the most important information in a table. A lot of information is duplicated.

Reply: The reviewer is right we should avoid duplicate information. Accordingly, we read carefully over the manuscript and made adjustments in this and other sections. 

For the section this reviewer is referring too, we aimed for a reasonable balance between details and redundancies while respecting a potentially wide readership. For PLOS One, we expect that some readers have a solid background in self-report questionnaires, and others are novices. Thus, for the latter, without detailed description, the nature of the questionnaire is difficult to grasp. They should not have to go to the original publications to know how such scales are constructed, but get all relevant information from reading the current manuscript. This would also include the sub-dimensions. It is also worth noting that we conducted two different studies in which we sometimes measured the same dimensions using two different questionnaires (e.g., Dark Triad). Thus, a detailed description helps the reader understand differences between studies and also resultant scores. 

R3.4.Table 1 : Cronbach 's alpha on the " psychopathy " scale is too low for analysis. Similarly, too low " Cronbach 's alpha " is on the in group trust level scale.

Reply: This comment is very welcome, because it forced us to think about this question, namely when low is “too low”. We went on a little journey to get reminded what these values actually imply. Does “too low” reliability also mean “too low” validity? Should we exclude the sub-scale scores all together? 

In our literature search, we found that low reliability does not necessarily impact predictive validity. Indeed, Cronbach’s alpha is a measure of internal consistency, but is of limited utility for homogeneity assessment (Clark & Watson, 1995). Calculating Cronbach’s alpha as an index of reliability allows us to determine the extent to which all the items of a scale measure the same construct. 

In more practical terms, Cronbach’s alpha is dependent on two parameters i) the number of tested items and ii) the average intercorrelations among them (Cronbach, 1951). Therefore, increasing the number of items will inevitably enhance internal consistency, while potentially decreasing validity due to the redundancy and added noise, leading to suboptimal construct assessment (see “the attenuation paradox” Loevinger, 1954). In our case, we assessed psychopathy using a validated self-report questionnaire (i.e., The French-Canadian Dirty Dozen; Zeng, 2013). The same was true for trust level (Inglehart et al., 2014). We agree that the reliability of these subscales was on the lower end (respectively, a Cronbach’s alpha of .596 for psychopathy and .510 for in group trust level) in our study. Yet, these alpha values fall within ranges of other published studies (e.g. Thalmayer & Saucier, 2011) and likely show that they measure related, yet not identical constructs. Worth noting, the psychopathy questionnaire we used in Study 2 had a higher Cronbach’s alpha than in Study 1 (.596 vs. 727), suggesting that the internal consistency was higher

R3.5. From verse 218 . Why is there so much information about the procedure and participants in the text itself? There is a diagram illustrating the procedure at the end, and the necessary information can be included in a table. This takes up a lot of space.

Reply: This comment echoes this reviewer’s comment R3.3. To avoid us repeating the same reply, we consider it best to refer back to a former reply (R.3.3). In addition, we also looked at the text to consider what we could delete in order to omit redundant information. Following the current reviewer’s comments as well as those of the other two referees, we have re-written the introduction and discussion alltogether, shortening and hopefully clarifying our reasoning and thoughts. 

R3.6.From verse 243 . Maybe I'm repeating myself, but why such a long description again when it can be included in a few sentences or a clear table?

Reply: Regarding the detailed descriptions, we invite the reviewer to read our previous response to R3.3. 

R3.7.Table 3 Should scales with such low reliability be included in the analysis (Psychopathy and in - group trust)?

Reply: Please see our reply to this reviewer’s comment R3.4. 

R3.8.From verse 292. Another test of students, and in psychology at that. Is this group representative in the self report?

Reply: We can understand that this issue comes up. We have touched upon the question of whether our sample is representative in our response to comment R3.2. In addition, we wish to add some words on the representativeness in comparison to “normative” values. Several of our scales were developed by testing undergraduate students. For example, to develop the cultural value scale (Zheng, 2013), the author tested 223 French students (104 males) from 18 to 35 years old. For the emotional intelligence scale, Maria and colleagues (2016) tested 824 French undergraduate students (368 males), with a mean age of 20.7 years (standard deviation of 2.1). Regarding the Short Dark Triad normative values (see Gamache et al., 2018), data were collected in a sample of 405 French-Canadian participants (with a mean age of 31.01, standard deviation of 11.97, age range 18-76 years). The latter sample was not exclusively composed of students, but the data were collected via institutional email from two universities. Only three out of our 22 predictors variables were tested in adult samples that were not student populations. For Empathy, Gilet and colleagues (2012) tested 322 French participants, with a mean age of 49.5 years old (standard deviation of 21.1, age rang 18-89 years). For social desirability, normative data were obtained from 1,159 French-Canadian participants (567 males) between 17 and 67 years old. For trust level, Delhey and colleagues (2012) tested 1,089 adults from 50 societies without providing age information. We conclude that most of our measurements are representative for our student population. 

R3.9.From verse 299 . Again, why are they introducing so many new research tools plus including the ones they used in the previous study. Do they want to measure everything in this article?

Reply: Reading this comment, we thought that this reviewer might experience some “desperation” given the number of possible individual differences measures we have been considering. And true, there are many possible ones to look at, due to a lack of research so far. Before COVID, we had launched Study 1, because we wanted to know about individual differences that might be worth pursuing when testing selected professional groups. Then COVID hit, we had our first results, and realized that some measures might have better psychometric alternatives. We presented the results of Study 1 to our peers and were advised to add a social desirability scale. Most important to us was the repetition of the study changing the self-reported lie detection abilities scale. We agree, we considered many variables, and obviously, given so many non-significant results, one might be tempted to only report a few of them. Yet, this would seem totally wrong given the widely discussed replication crises (Shrout & Rodgers, 2018). We feel very committed to report everything we have done. Therefore, we are both transparent and informative to peers who might consider the same individual difference measures. As Dr. Curtis, reviewer 2, stated “the results are what they are”. 

R3.10.Table 4 . Again, we have quite poor reliability on several scales: in trust group, empathic concerns , perspective taking , general score of social desirability .

Reply: Please see our response to comment R3.4. 

R3.11.From verse 338. Do you really need to re-detail what is contained in the table?

Reply: Please see our response to comment R3.3 above. 

R3.12.Verse 406. Isn't 21 self -report variables too much for one article?

Reply: Please see our response to comment R3.9 above.

R.3.13.Verse 415. They put in so many variables and only 13 percent of the variance is explained? It's probably a very small number. Maybe a bad theory?

Reply: This reviewer is coming back with very similar remarks. We are uncertain what else we can explain. Sure, the reviewer and other potential readers can question our study. Yet, we wish to reiterate that there is very little research on this self-report bias. There is no solid theory yet. We need to describe a phenomenon first, and we are dedicated to this effort. This self-report bias could have real impact on human decision making (being convinced that somebody is lying, because one believes being able to spot lying). Once we have described this self-report bias and its correlates, we feel ready to also consider theories. This approach might appear to some unscientific, and also to some who teach scientific methods. However, recent voices highlight the need to allocate more time and efforts in the description of any type of phenomena, before theories are presented, and mechanisms are investigated (Scheel et al., 2021). Regarding the explained variance of the model, to encompass the fact that adding more variables increases the amount of explained variance, we rely on the adjusted R-squared which has been adjusted for the number of the predictors in the model and does not simply increase with more predictors (unlike unadjusted R-squared).

R3.14. Verse 442. Of the 21 variables, only 2 are significant predictors. 

Reply: We are not sure what the reviewer wishes us to comment on, or change. Thus, we consider it possible that they feel that we report on many variables that are not significant, which they mentioned also in other comments (see also our reply to comment R3.9 and R3.13). Thus, we wish the referee to read our response there, else, we would to receive more specific information what we should do.

R3.15.Verse 447. Since such a small number of significant variables was unexpected, perhaps it was better to start the theories earlier instead of coming to such conclusions only now after a huge amount of work.

Reply: Please see our response to comment R3.13.

R3.16.Verse 483-485. I don't understand this line of reasoning. On what theoretical basis is such a conclusion?

Reply: Most of the studies on self-reported lie detection abilities and the Dark Triad found significant relationships (e.g. Wissing & Reinhard, 2017; Zvi & Elaad, 2018;Elaad et al., 2020; Elaad 2022). We did not. Thus, we detailed why these differences might have emerged between studies. As announced for Study 2, we added a second measure of self-reported lie detection abilities, one that compares own abilities to other persons’ abilities. We had been interested in what people think in principle about their abilities, not in comparison to others. Thus, in Study 2, we assessed both self-reported measures. In line with reviewer 1 (see our response to comment R1.3), we now report results on both measures. We also discuss the results in the revised manuscript on page 22-24 

The verse 483-485 refers to results reported in Wissing & Reinhard (2017), namely a significant link between psychopathy and self-reported lie detection abilities using the added measurement in Study 2 (self-reported lie detection abilities in comparison to others). The way these authors assessed self-reported lie detection abilities differs from the measure we used in Study 1 and again in Study 2 (i.e., being able to detect lies in general). 

R3.17.Verse 529. Instead of a larger sample of respondents, maybe it's better not to study only psychology students?

Reply: see our reply to comment R3.2 and R3.8.

R3.18.Overall, there is a lot of chaos in this article and it lacks a cohesive structure. The authors tackle an interesting problem, but they are actually lost in the large amount of research and variables introduced that don't really explain the phenomenon of detecting lies in other people. After reading this article, it is basically unknown what these individual differences in the title would consist of.

The article is too long-winded. It seems like it could be half as long. It contains numerous repetitions and doubles. In the theoretical part, the authors briefly cite a lot of variables and studies related to lie detection, the description of which, instead of explaining this phenomenon, only complicates its understanding. In addition, in the theoretical part, they mix theory with the method that should be in the next part. 

Reply: see reply to comments R3.3, R3.9. We can only add here that we read and reworked the manuscript having comments of three referees in mind.

R3.19.Again, there is the problem of text systematization and structure

In the methodological part, the selection of samples is puzzling. Maybe students (including psychology) are not necessarily a representative group for such a common phenomenon as lying and its detection. Then there is the issue of a large number of questionnaires and variables for one article.

Reply: Please see our reply to comments R3.2, R3.3, R3.8 and R3.9.

R3.20.The article contains numerous mental abbreviations that are not always understandable to the reader.

Reply: The mental abbreviations used in our article are always preceded by both the spelled-out version and the short form. Following the reviewer’s comment, we decided to remove all the abbreviations and corrected the manuscript accordingly.

R3.21.The structure of sentences, which is sometimes too complicated, does not help to understand the article. In conclusion, it seems that the authors put a lot of work into this article, but it is too vague for the reader.

Reply: We hope that our extensive re-write improved the simplicity of the manuscript.

R3.22.The manuscript deals with an interesting topic, but in the manuscript too often the names can refer to "lie detection" which the authors have not studied. "lie detection" is not possible to examine by self-report. 

Reply: We have done a careful reading through the manuscript keeping this comment in mind. We have been careful that we always stated when studies looked at lie detection (and did so methodologically) or tested the belief that one is able to detect lying (no actual lie detection was assessed). We were also careful in our phrasing for the very few studies that looked at self-reported lie detection abilities and the actual assessment of lie detection abilities.

R3.23.It will also be advisable to reduce the number of variables to only those that have theoretical justification. The revised text will certainly be very interesting.

Reply: From the previous comments by this reviewer, we understand that they would prefer us to exclude information and data we have collected. Yet, this would be unacceptable to us, given the extensive discussion on replication problems, publication biases, data spotting, data polishing, that has brought a dark shadow on many empirical research fields, not only on psychology. Zero results should be welcomed, and all the collected data and measures should be reported. In this regard, we cannot follow this suggestion.

 

References not mentioned in the manuscript:

Clark LA, & Watson DB. (1995). Constructing validity: basic issues in objective scale development. Psychological Assessment, 7(3), 309-319.

Cohen, J. (1988). Statistical power analysis for the behavioral. Sciences. Hillsdale (NJ): Lawrence Erlbaum Associates, 18, 74.

Cohen, J. (1992). Statistical power analysis. Current directions in psychological science, 1(3), 98-101. 

Cronbach, L.J. Coefficient alpha and the internal structure of tests. Psychometrika 16, 297–334 (1951). https://doi.org/10.1007/BF02310555

Henrich, J., Heine, S. J., & Norenzayan, A. (2010). Most people are not WEIRD. Nature, 466(7302), 29-29.

Holden RR, Passey J. (2010). Socially desirable responding in personality assessment: Not necessarily faking and not necessarily substance. Personality and Individual Differences, 49(5): 446-450. 

Loevinger J. (1954). The attenutation paradox in test theory. Psychological Bulletin, 51(5), 493-504.

Saucier, G. (2009). Recurrent personality dimensions in inclusive lexical studies: Indications for a Big Six structure. Journal of personality, 77(5), 1577-1614.

Scheel, A. M., Tiokhin, L., & Isager, P. M. (2021). Why Hypothesis Testers Should Spend Less Time Testing Hypotheses. Perspectives on Psychological Science, 16(4), 744-755. 

Shrout, P.E., & Rodgers, J.L. (2018). Psychology, science, and knowledge construction: Broadening perspectives from the replication crisis. Annual review of psychology, 69, 487-510. 

Thalmayer, A. G., & Saucier, G. (2014). The questionnaire big six in 26 nations: Developing cross–culturally applicable big six, big five and big two inventories. European Journal of Personality, 28(5), 482-496.

Thalmayer, A. G., Saucier, G., & Eigenhuis, A. (2011). Comparative validity of brief to medium-length Big Five and Big Six Personality Questionnaires. Psychological assessment, 23(4), 995.

Thalmayer, A. G., & Saucier, G. (2014). The questionnaire big six in 26 nations: Developing cross–culturally applicable big six, big five and big two inventories. European Journal of Personality, 28(5), 482-496.

---

## [Decision Letter · Decision Letter 1]

16 Apr 2023

Individual differences in self-reported lie detection abilities.

PONE-D-22-29409R1

Dear Dr. Fernandes,

We’re pleased to inform you that your manuscript has been judged scientifically suitable for publication and will be formally accepted for publication once it meets all outstanding technical requirements.

Kind regards,

Peter Karl Jonason

Academic Editor

PLOS ONE

Additional Editor Comments (optional):

Reviewers' comments:

Reviewer's Responses to Questions

**Comments to the Author**

1. If the authors have adequately addressed your comments raised in a previous round of review and you feel that this manuscript is now acceptable for publication, you may indicate that here to bypass the “Comments to the Author” section, enter your conflict of interest statement in the “Confidential to Editor” section, and submit your "Accept" recommendation.

Reviewer #2: All comments have been addressed

2. Is the manuscript technically sound, and do the data support the conclusions?

Reviewer #2: Yes

3. Has the statistical analysis been performed appropriately and rigorously? 

Reviewer #2: I Don't Know

4. Have the authors made all data underlying the findings in their manuscript fully available?

Reviewer #2: No

5. Is the manuscript presented in an intelligible fashion and written in standard English?

Reviewer #2: Yes

6. Review Comments to the Author

Reviewer #2: The authors have addressed my comments on the original paper and done a good job of addressing the comments of the other revierwers. My only hesitation is that the Supporting/supplementary materials were not avalable to me as a review and I was interested to see the tables because, without them, the results of Study 1 are very sparce.

7. PLOS authors have the option to publish the peer review history of their article (what does this mean?). If published, this will include your full peer review and any attached files.

Reviewer #2: **Yes: **Guy J. Curtis

---

## [Editor Report · Acceptance letter]

27 Apr 2023

PONE-D-22-29409R1 

Individual differences in self-reported lie detection abilities. 

Dear Dr. Fernandes:

I'm pleased to inform you that your manuscript has been deemed suitable for publication in PLOS ONE. Congratulations! Your manuscript is now with our production department. 

Kind regards, 

on behalf of

Dr. Peter Karl Jonason 

Academic Editor

PLOS ONE